# Parkinson Disease Protein 7 (PARK7) Is Related to the Ability of Mammalian Sperm to Undergo In Vitro Capacitation

**DOI:** 10.3390/ijms221910804

**Published:** 2021-10-06

**Authors:** Sandra Recuero, Ariadna Delgado-Bermúdez, Yentel Mateo-Otero, Estela Garcia-Bonavila, Marc Llavanera, Marc Yeste

**Affiliations:** 1Biotechnology of Animal and Human Reproduction (TechnoSperm), Institute of Food and Agricultural Technology, University of Girona, ES-17003 Girona, Spain; sandra.recuero@udg.edu (S.R.); ariadna.delgado@udg.edu (A.D.-B.); yentel.mateo@udg.edu (Y.M.-O.); estela.garcia@udg.edu (E.G.-B.); marc.llavanera@udg.edu (M.L.); 2Unit of Cell Biology, Department of Biology, Faculty of Sciences, University of Girona, ES-17003 Girona, Spain

**Keywords:** sperm, in vitro capacitation, PARK7, pig, ROS levels

## Abstract

Parkinson disease protein 7 (PARK7) is a multifunctional protein known to be involved in the regulation of sperm motility, mitochondrial function, and oxidative stress response in mammalian sperm. While ROS generation is needed to activate the downstream signaling pathways required for sperm to undergo capacitation, oxidative stress has detrimental effects for sperm cells and a precise balance between ROS levels and antioxidant activity is needed. Considering the putative antioxidant role of PARK7, the present work sought to determine whether this protein is related to the sperm ability to withstand in vitro capacitation. To this end, and using the pig as a model, semen samples were incubated in capacitation medium for 300 min; the acrosomal exocytosis was triggered by the addition of progesterone after 240 min of incubation. At each relevant time point (0, 120, 240, 250, and 300 min), sperm motility, acrosome and plasma membrane integrity, membrane lipid disorder, mitochondrial membrane potential, intracellular calcium and ROS were evaluated. In addition, localization and protein levels of PARK7 were also assessed through immunofluorescence and immunoblotting. Based on the relative content of PARK7, two groups of samples were set. As early as 120 min of incubation, sperm samples with larger PARK7 content showed higher percentages of viable and acrosome-intact sperm, lipid disorder and superoxide levels, and lower intracellular calcium levels when compared to sperm samples with lower PARK7. These data suggest that PARK7 could play a role in preventing sperm from undergoing premature capacitation, maintaining sperm viability and providing a better ability to keep ROS homeostasis, which is needed to elicit sperm capacitation. Further studies are required to elucidate the antioxidant properties of PARK7 during in vitro capacitation and acrosomal exocytosis of mammalian sperm, and the relationship between PARK7 and sperm motility.

## 1. Introduction

Despite being mature and motile once ejaculated, mammalian sperm must reside in the female reproductive tract to acquire their fertilizing ability, in a process known as capacitation [1,2,3]. During capacitation, sperm undergo an assortment of physiological and biochemical changes, including alterations in plasma membrane permeability and fluidity [4,5], acrosome integrity [6], intracellular pH and calcium levels [7,8], mitochondrial activity [9], and the generation of reactive oxygen species (ROS) [10]. Oxidative stress, caused by excessive ROS formation, has detrimental consequences for sperm cells and fertility, including a decrease in sperm motility and their ability to interact with the oocyte, an increase in plasma membrane lipid peroxidation, protein damage and DNA fragmentation, and even the activation of apoptotic-like pathways [11]. Notwithstanding, moderate levels of ROS are needed to perform some physiological processes, such as sperm capacitation [10,12]. Indeed, ROS generation during early capacitation events induces a rise of cAMP levels and the activation of protein kinases, which, in turn, trigger the phosphorylation of tyrosine residues of certain proteins, stimulating specific downstream pathways [13,14]. Thus, proper ROS homeostasis is required to avoid sperm damage.

Parkinson disease protein 7 (PARK7) or protein deglycase DJ-1 is a multifunctional protein encoded by the *PARK7* gene in chromosome 6, which has been found to be highly expressed in a large variety of mammalian tissues [15]. Although its biochemical function in sperm has not been fully studied, PARK7 and its rat homolog SP22/CAP1 have been reported to be involved in spermatogenesis [16,17], fertilization [18], motility [19], mitochondrial function [20], and oxidative stress response [20,21]. In human sperm, PARK7 is localized in the surface of the sperm head, the anterior part of the midpiece, and in the principal piece, reinforcing its putative role for oocyte binding during fertilization and for flagellar movement [22]. Similarly, PARK7 has also been detected on the surface of the equatorial segment of sheep [17], horse [23], rat, hamster, rabbit, and cattle sperm [18]. However, the precise localization of PARK7 in pig sperm is yet to be determined. With regard to the response to oxidative stress, PARK7 has been positively correlated with variables affected by ROS generation in human sperm, such as membrane integrity [17], total motility and superoxide dismutase (SOD) activity [19]. In patients with asthenozoospermia, characterized by a reduction in sperm motility, the relative content of PARK7 is lower in both ejaculated sperm [19,24] and seminal plasma [25]. In addition, PARK7 has been identified as a target of miR-4485-3, whose downregulation has also been associated to asthenozoospermia [26]. Under oxidative stress conditions associated with asthenozoospermia, PARK7 has been observed to be translocated from the equatorial segment to the midpiece, which has been proposed as a possible protective mechanism for sperm to maintain their mitochondrial function [20]. In addition to men suffering from asthenozoospermia, PARK7 is downregulated in patients with varicocele [27] and oligospermia [28], thus emphasizing its importance for male fertility.

Sperm handling and storage, such as cryopreservation or liquid conservation, have detrimental effects on sperm cells, including a reduction in viability and plasma membrane integrity, alterations in motility and mitochondrial function, and generation of oxidative stress [29,30]. Some of these perturbations share similarities with the events that take place during sperm capacitation and, for this reason, some authors refer to them as capacitation-like changes [31]. Besides that stated before, it is well known that changes in the levels, localization or function of specific proteins also occur [29]; for example, PARK7 is lost during liquid storage and cryopreservation of equine [23] and porcine sperm [32]. However, and to the best of our knowledge, whether or not PARK7 is involved in sperm capacitation has not been investigated in any species. Considering all the aforementioned, the present study sought to determine if PARK7 is related to the sperm ability to undergo in vitro capacitation and trigger the acrosomal exocytosis induced by progesterone using the pig as a model. Specifically, changes in the localization of this protein during in vitro capacitation were overseen, and whether the relative content of PARK7 in fresh sperm is related to their ability to withstand in vitro capacitation was interrogated. 

## 2. Results

### 2.1. Classification of Sperm Samples Based on Their Relative PARK7 Content

Based on the relative content of PARK7 (PARK7/α-tubulin ratio), determined using immunoblotting analysis at the beginning of the experiment (0 min), two different groups of sperm samples, with high (*n* = 4; 1.55 ± 0.56; mean ± SD) and low relative levels of PARK7 (*n* = 4; 0.45 ± 0.13), were distinguished. The sperm samples with high relative PARK7 content at 0 min showed significantly (*p* < 0.05) higher levels of PARK7 than the group with low relative PARK7 content in all the time points assessed, except at 240 min when no statistical differences were observed (Figure 1). As shown in Figure 1, PARK7 levels decreased along incubation in capacitation medium, especially in sperm with high relative PARK7 content. Figure 2a,b show representative blots for α-tubulin and PARK7, respectively. The peptide competition assay for the anti-PARK7 antibody demonstrated that the band at 25 kDa was specific for PARK7 (Figure 2c). 

### 2.2. Sperm Motility

Regarding the total (Figure 3a) and progressive (Figure 3b) motility, no significant differences (*p* > 0.05) between the sperm samples with high and low relative PARK7 levels were observed during in vitro capacitation and progesterone-induced acrosomal exocytosis (300 min). Although no significant differences were observed between the different time points in the group of samples with low relative PARK7, total motility also tended to decrease along incubation (0 min vs. 300 min; *p* = 0.060).

In relation to kinetic parameters, shown in Table 1, curvilinear velocity (VCL, μm/s) presented a tendency to be higher in samples with high PARK7 compared to those with low PARK7 after 250 min of incubation (*p* = 0.052). The same tendency was observed regarding straight-line velocity (VSL, μm/s; *p* = 0.061) and linearity (LIN, %; *p* = 0.068). Whereas the amplitude of lateral head displacement (ALH, μm) was significantly higher in samples with high relative PARK7 content than in those with low levels of this protein at 240 min (*p* < 0.05), straightness (STR, %) was significantly higher in the sperm with high relative amount of PARK7 compared to those with low PARK7 after 250 min of incubation (*p* < 0.05).

### 2.3. Sperm Viability

The percentage of viable sperm (i.e., sperm with intact plasma membrane, SYBR14^+^/PI^−^) was similar between the groups along in vitro capacitation and progesterone-induced acrosomal exocytosis (Figure 4a). However, the viability of sperm samples with high relative PARK7 content presented a clear tendency to be higher compared to those with low relative PARK7 levels after 120 min of incubation (*p* = 0.057). On the other hand, although no significant differences in sperm viability between the different time points were observed in the group of samples with low relative amounts of PARK7, the percentage of viable sperm was significantly lower (*p* < 0.05) after 120 min of incubation than at the beginning of the experiment (0 min).

### 2.4. Membrane Lipid Disorder

After 120 min of incubation in capacitation medium, the percentage of viable sperm with high lipid membrane disorder (M540^+^/YO-PRO^−^) was higher in the sperm samples with high relative PARK7 content than in those with lower levels of this protein (*p* < 0.05) (Figure 4b). Within the group of samples with high relative amounts of PARK7, the percentage of viable sperm with high membrane lipid disorder significantly decreased after 300 min of incubation compared to 120 min (*p* < 0.05). Regarding the population of viable sperm with low lipid membrane disorder (M540^−^/YO-PRO^−^), the percentage was higher in the sperm samples with high relative PARK7 content than in those with low relative PARK7 levels (*p* < 0.05) after 120 min of incubation (Appendix A). As far as the population of non-viable sperm with high membrane lipid disorder (M540^+^/YO-PRO^+^) is concerned, the percentage was higher in the sperm samples with low relative PARK7 content compared to those with high relative PARK7 amounts after 120 and 240 min of incubation (*p* < 0.05) (Appendix A).

### 2.5. Acrosome Membrane Integrity

With regard to the integrity of the acrosome membrane, the percentage of viable sperm with an intact acrosome membrane (PNA-FITC^−^/PI^−^) was higher in the samples with high relative PARK7 content than in those with low relative PARK7 levels after 120 and 240 min of incubation (*p* < 0.05) (Figure 5a). Conversely, the percentage of non-viable sperm with an outer acrosome membrane that could not be fully intact (PNA-FITC^+^/PI^+^) was significantly lower in the sperm with high relative PARK7 content than in those with low relative PARK7 levels after 120 and 240 min of incubation (*p* < 0.05) (Appendix A). In samples with low relative amounts of PARK7, this percentage exhibited a prompt increase at 120 min compared to the beginning of the experiment. This increase was maintained until the end of the incubation in a capacitation medium (300 min).

### 2.6. Intracellular Calcium Levels

The percentage of viable sperm with low intracellular calcium levels (Fluo3^−^/PI^−^) was significantly higher in the samples with high relative PARK7 content compared to those with low relative PARK7 after 120 min of incubation in capacitation medium (*p* < 0.05) (Figure 5b). Although this difference was not statistically significant, this percentage presented a tendency to be higher in the sperm with high relative amounts of PARK7 than in those with low relative PARK7 levels after 300 min of incubation (*p* = 0.056). 

### 2.7. Intracellular ROS Levels

After 120 and 240 min of incubation in capacitation medium, the percentage of viable sperm with high O_2_^−^ levels (E^+^/YO-PRO-1^−^) was significantly higher in the sperm samples with high relative PARK7 content than in those with low relative PARK7 levels (*p* < 0.05) (Appendix A). Considering the viable sperm population only, the percentage of cells with high intracellular superoxide levels (E^+^/viable sperm population) was found to be significantly higher in the sperm with high than in those with low relative PARK7 content after 120 min of incubation (*p* < 0.05) (Figure 6a). In contrast, no significant differences between the samples with high and low relative PARK7 levels were found regarding intracellular peroxide levels during in vitro capacitation and progesterone-induced acrosomal exocytosis (*p* > 0.05). 

### 2.8. Mitochondrial Membrane Potential (MMP)

The differences in the initial relative levels of PARK7 in the sperm did not affect MMP during in vitro capacitation and progesterone-induced acrosomal exocytosis. However, although not significant, the percentage of sperm exhibiting high MMP tended to be lower in sperm samples with high relative PARK7 content than in those with low relative levels of this protein after 300 min of incubation in capacitation medium (*p* = 0.063) (Figure 6b).

### 2.9. Localization of PARK7 in Sperm during In Vitro Capacitation and Acrosomal Exocytosis

Localization of PARK7 protein in sperm was assessed through immunofluorescence. As shown in Figure 7, green fluorescence was detected in the sperm flagellum, midpiece, the anterior part of the head and the post-acrosomal region. After antibody specificity assay (Figure 8), fluorescence staining of the flagellum and the post-acrosomal region disappeared, which confirmed these two regions as specific localizations of PARK7. On the other hand, PARK7-immunolocalization was also evaluated at each relevant time point (0, 120, 240, 250 and 300 min) (Figure 7) in order to determine whether PARK7 relocalized during in vitro capacitation and acrosomal exocytosis. As shown in Figure 7, the localization pattern of PARK7 in sperm was maintained along the incubation in capacitation medium (300 min).

## 3. Discussion

During the physiological process through which a spermatozoon acquires the ability to penetrate the zona pellucida and fuse with the oocyte, namely capacitation, the sperm cells undergo a variety of changes including modifications in the fluidity of plasma and acrosome membranes [4,5,6], increase in intracellular calcium and ROS levels [7,8,10], and tyrosine phosphorylation of certain proteins [33,34]. In fact, the ROS generated during early capacitation events are related to calcium compartmentalization [35] and tyrosine phosphorylation [10], which are both required for triggering the downstream signaling pathways that lead sperm to achieve the capacitated status. In this regard, a balance between ROS formation and antioxidant activity is essential to avoid the harmful effects caused by oxidative stress. 

Although PARK7 protein has been extensively studied in the context of neurodegenerative disorders (reviewed by [36]), it is known to be expressed not only in the brain but also in other tissues and organs, including male epididymis and testis [22]. Although its precise function in sperm is not fully understood, PARK7 is associated with the oxidative stress response [20,21]. In humans, PARK7 has been positively correlated with sperm plasma membrane integrity [17], total motility and SOD activity [19]. Additionally, the relevance of PARK7 for male fertility has been previously investigated. In effect, PARK7 has been found downregulated in patients with asthenozoospermia [19,24], varicocele [27], or oligozoospermia [28]. To the best of our knowledge, however, whether or not PARK7 is related to the sperm ability to undergo in vitro capacitation and trigger the progesterone-induced acrosomal exocytosis has not been investigated in any mammalian species, including humans. The present study demonstrated, for the first time and using the pig as an animal model, that the ability of mammalian sperm to undergo in vitro capacitation is related to their relative PARK7 content. Herein, the relative content of PARK7 was shown to decrease along the incubation of sperm in capacitation medium. This is in agreement with an earlier study that also found that the relative PARK7 content decreases after capacitation of pig sperm [37]. Moreover, the statistical approach performed in this work allowed distinguishing two different groups of sperm samples, based on the relative levels of PARK7 at the beginning of the experiment (0 min). Following this, the parameters associated with sperm capacitation were compared between these two groups, and how such potential changes were influenced by the relative PARK7 content was interrogated. Moreover, the localization pattern of PARK7 in pig sperm was not previously investigated, nor were the potential changes in that localization during in vitro capacitation and acrosomal exocytosis in any mammalian species addressed. The immunofluorescence assay revealed that the localization pattern of PARK7 is maintained during in vitro capacitation and acrosomal exocytosis and that this protein is located in the post-acrosomal region and tail of pig sperm. Similarly, in humans, PARK7 is present on the surface of the posterior part of the head, the midpiece and the flagellum [22].

Regarding the integrity of sperm plasma membrane, clear differences between samples with high and low PARK7 levels were found. Samples with low relative PARK7 content at the onset of the experiment experienced a severe decline in sperm viability during the first 120 min of incubation. In addition, non-viable sperm with high membrane lipid disorder were significantly higher in the group with low PARK7 levels. This high asymmetry in membrane lipids is likely to induce the destabilization of the plasma membrane, leading to sperm death. Remarkably, while a decrease in membrane integrity and, therefore, sperm viability, was expected during in vitro capacitation [38,39], such a reduction occurred more progressively in sperm samples that contained higher PARK7 levels. As mentioned earlier, PARK7 levels have been positively correlated with plasma membrane integrity [17]. Thus, in mammalian sperm that present a plasma membrane particularly sensitive to oxidative stress due to its polyunsaturated fatty acid composition [40], PARK7 could maintain membrane integrity through its antioxidant function. Along these lines, the integrity of the acrosome membrane was also highly altered in sperm samples with low relative PARK7 content at the beginning of the experiment. Whereas sperm samples with high PARK7 levels maintained their acrosome intact during the first 240 min of incubation, those with low PARK7 presented a significant decline in the integrity of both plasma and acrosome membrane. One possible explanation is that such a loss of acrosome integrity was a direct consequence of the aforementioned plasma membrane destabilization. In addition, in samples with low PARK7 content, viable sperm with low intracellular calcium levels promptly decreased, concomitantly with the loss of acrosomal and plasma membrane integrity during the first 120 min of incubation. Bearing in mind that intracellular calcium is essential to trigger acrosomal exocytosis [41] and considering that membrane destabilization could lead to alterations in ion permeability, an increase in calcium influx could provoke a premature acrosome exocytosis. 

As pointed out earlier, high amounts of ROS have detrimental effects on sperm cells and, thus, a precise control of their levels is necessary during sperm capacitation. It is known that antioxidant molecules have the potential to modulate the capacitation status of sperm [35,42]. The antioxidant enzyme glutathione peroxidase 6 (GPX6) has been posited to prevent sperm from premature capacitation by removing the excess of ROS [43]. Similarly, another enzyme with antioxidant properties, paraoxonase 1 (PON-1), can hinder sperm from spontaneous acrosome reaction [44]. In this sense, PARK7 could also play a role modulating ROS levels during sperm capacitation. Strikingly, superoxide levels were higher in samples containing high relative PARK7 content; those samples also showed higher viability than the ones with low levels of PARK7. Furthermore, in humans, concentration of PARK7 in ejaculated sperm has been positively correlated with SOD activity, which catalyzes the conversion of superoxides into hydrogen peroxide [19]. Considering that sperm with high PARK7 content showed higher superoxide levels and that no significant differences were observed with regard to peroxide levels, PARK7 does not seem to be associated with SOD activity during in vitro capacitation. Therefore, the data obtained herein suggest that PARK7 could maintain sperm viability when superoxide levels are elevated, thus providing sperm with a higher tolerance to ROS generation needed for sperm capacitation, among other physiological processes. 

Finally, and regarding motility, it has been proposed that mammalian sperm, including the porcine, display a specific motility pattern under capacitation conditions [45]. In humans, PARK7 levels have been positively correlated with sperm motility [19]. In spite of this, in the present study significant differences in total or progressive motility were not observed. Notwithstanding, significant differences or tendencies were observed in several kinematic sperm parameters, including VCL, VSL, LIN, ALH, and STR. Thus, it is reasonable to reckon that PARK7 plays a role in the regulation of sperm motility during capacitation. Taking this into consideration, future studies including more animals are needed to elucidate the relation of PARK7 with motility regulation in porcine sperm.

## 4. Materials and Methods

Unless otherwise stated, all reagents used in the present study were purchased from Sigma-Aldrich (Saint Louis, MO, USA).

### 4.1. Semen Samples

Ejaculates from 32 different sexually mature Piétrain boars were provided by an authorized artificial insemination (AI) center (Grup Gepork S.L., Masies de Roda, Spain). Boars were fed with a standard and balanced diet with water being provided ad libitum. According to the farm records, all boars were fertile. Ejaculates were manually collected through the gloved-hand method. The sperm-rich fraction was diluted in a commercial extender for liquid storage (Vitasem LD; Magapor S.L., Zaragoza, Spain) at a final concentration of 3.3 × 10^7^ sperm/mL. Seminal doses of 90 mL and 3 billion sperm per dose were transported at 17 °C to the laboratory within three hours post-collection. 

All procedures involving animals were performed according to the EU Directive 2010/63/EU for animal experiments, the Animal Welfare Law and the current regulation on Health and Biosafety issued by the Department of Agriculture, Livestock, Food and Fisheries (Regional Government of Catalonia, Spain). Production of seminal doses by the AI center followed the ISO certification (ISO-9001:2008). As seminal doses were directly purchased from the local AI center, the authors did not manipulate any animal and no approval from an ethics committee was required.

### 4.2. In Vitro Capacitation and Progesterone-Induced Acrosomal Exocytosis

Four seminal doses from different boars were randomly pooled in each experiment (*n* = 8 different experiments; 32 boars), in order to avoid variability between males. Pooled semen samples were washed with PBS (centrifugation at 600× *g* and 17 °C for 5 min) as indicated by Yeste et al. [8], and pellets were finally resuspended in 40 mL of capacitation medium (CM) at a final concentration of 2.7 × 10^7^ sperm/mL. The CM consisted of 20 mM HEPES (20 mM 4-(2-hydroxyethyl)-1-piperazineethanesulfonic acid), 130 mM NaCl, 3.1 mM KCl, 5 mM glucose, 21.7 mM sodium L-lactate, 1 mM sodium pyruvate, 0.3 mM Na_2_HPO_4_, 0.4 mM MgSO_4_·7H_2_O, 4.5 mM CaCl_2_·2H_2_O, 15 mM NaHCO_3_, and 5 mg/mL of bovine serum albumin (BSA). The osmolality was checked to be in the range of 290–310 mOsm/kg, and the pH was adjusted to 7.4. Sperm were incubated at 38 °C and 5% CO_2_ for 300 min (Heracell 150; Heraeus Instruments GmbH, Osterode, Germany). After 240 min of incubation, progesterone was added at a final concentration of 10 μg/mL, in order to induce the acrosomal exocytosis [46,47].

At each relevant time point (0, 120, 240, 250, and 300 min), which were set based on previous studies [8,9], the following sperm parameters were evaluated: sperm viability, membrane lipid disorder, acrosome integrity, mitochondrial membrane potential, and intracellular levels of calcium, peroxides, and superoxides. In addition, two different aliquots were taken for immunoblotting and immunofluorescence analysis. Samples for protein extraction were washed twice (PBS, pH = 7.3) at 300× *g* for 5 min (room temperature) to remove the CM; sperm pellets were stored at −80 °C for further processing. 

### 4.3. Sperm Motility

Evaluation of total and progressive sperm motility was performed using a commercial computer assisted sperm analysis (CASA) system. This system consisted of a phase-contrast microscope (Olympus BX41; Olympus, Hamburg, Germany) connected to a computer equipped with the Integrated Semen Analysis System (ISAS^®^) V1.0 software (Proiser S.L., Valencia, Spain). Briefly, 5 μL of each semen sample was placed onto a pre-warmed (37 °C) Makler counting chamber (Sefi Medical Instruments, Haifa, Israel). Then, samples were observed under a negative phase-contrast objective at 100× magnification (Olympus 10× 0.30 PLAN objective lens). Three replicates, with a minimum of 500 sperm per replicate, were counted in each sample per time point. The corresponding mean ± standard deviation (SD) was calculated. The sperm motility parameters recorded using the software were the following: total motility (TMOT, %); progressive motility (PMOT, %); curvilinear velocity (VCL, μm/s); average path velocity (VAP, μm/s); straight-line velocity (VSL, μm/s); amplitude of lateral head displacement (ALH, μm); beat cross frequency (BCF, Hz); linearity (LIN, %); wobble coefficient (WOB, %); and straightness (STR, %). Total motility was defined as the percentage of sperm showing VAP ≥ 10 μm/s, and progressive motility was defined as the percentage of motile sperm showing STR ≥ 45%. These parameters are described in more detail in Verstegen et al. [48].

### 4.4. Flow Cytometry Analyses

The following seven sperm variables were evaluated using flow cytometry at each relevant time point: sperm viability (i.e., plasma membrane integrity), acrosome integrity, membrane lipid disorder, acrosome membrane integrity, and intracellular levels of superoxides (O_2_^−^), peroxides (H_2_O_2_), and calcium. Samples were diluted in PBS (pH = 7.3) to a final concentration of 2 × 10^6^ sperm/mL prior to fluorochrome staining. Unless otherwise stated, all fluorochromes were acquired from ThermoFisher Scientific (Waltham, MA, USA). Flow cytometry analyses were conducted using a Cell Laboratory QuantaSC cytometer (Beckman Coulter, Fullerton, CA, USA). Samples were excited with an argon ion laser (488 nm) set at power of 22 mW, and laser voltage and rate were constant along the experiment. For each event, the cytometer provided the electronic volume (EV) and side scatter (SS). Three different optical filters (FL-1, FL-2, and FL-3) were used with the following optical properties: FL-1: Dichroic/Splitter (DRLP): 550 nm, BP filter: 525 nm, detection width 505–545 nm. FL-1 detected green fluorescence (SYBR-14; YO-PRO-1; 5,5′,6,6′-tetrachloro-1,1′,3,3′tetraethyl-benzimidazolylcarbocyanine iodide monomers, JC-1 monomers, JC-1_mon_; peanut agglutinin conjugated with fluorescein isothiocyanate, PNA-FITC; 2′,7′-dichlorofluorescin diacetate, H_2_DCFDA; and Fluo-3-AM). FL-2: DRLP: 600 nm, BP filter: 575 nm, detection width: 560–590 nm. FL-2 allowed detecting orange fluorescence (JC-1 aggregates, JC-1_agg_), and FL-3: LP filter: 670, detection width: 670 ± 30 nm. FL-3 was used to collect red fluorescence (merocyanine 540, M540; hydroethidine, HE; and propidium iodide, PI). For each staining protocol, photomultiplier settings were adjusted, and signals were logarithmically amplified. The analyzer threshold was adjusted on the EV channel to exclude subcellular debris (particle diameter < 7 μm) and cell aggregates (particle diameter >12 μm).

Flow rate was set at 4.17 μL/min in all analyses and three technical replicates, with a minimum of 10,000 events per replicate, were evaluated for each sample and sperm parameter. Flow cytometry data analysis was performed using Flowing Software (Ver. 2.5.1; Turku Bioscience, Turku, Finland), according to the recommendations of the International Society for Advancement of Cytometry (ISAC). Following this, the corresponding mean ± SD was calculated. 

#### 4.4.1. Sperm Viability (SYBR-14/PI)

Evaluation of sperm viability was performed by assessing plasma membrane integrity, using the LIVE/DEAD Sperm Viability Kit (Molecular Probes, Eugene, OR, USA), following the protocol set by Garner and Johnson [49]. In brief, sperm were stained with SYBR-14 (100 nmol/L) for 10 min at 38 °C in the dark, and next, with PI (12 μmol/L) for 5 min at the same conditions. Flow cytometry dot-plots were generated based on the combination of SYBR-14 and PI, resulting in the following three sperm populations: (i) viable, green-stained sperm (SYBR-14^+/^PI^−^); (ii) non-viable, red-stained sperm (SYBR-14^−^/PI^+^); and (iii) non-viable sperm stained in both green and red (SYBR-14^+^/PI^+^). SYBR-14 spill over into the FL-3 channel was compensated (2.45%). The percentage of non-sperm particles, corresponding to non-stained debris particles (SYBR-14^−^/PI^−^), was used to correct the percentage of particles within the double negative quadrant of every described parameter. The percentages of the other three populations were also recalculated. This percentage of non-sperm particles was also used to correct the data from the other tests.

#### 4.4.2. Membrane Lipid Disorder (M540/YO-PRO-1)

The assessment of membrane lipid disorder was performed using M540 and YO-PRO-1 fluorochromes, based on the protocol from Rathi et al. [50], as modified by Yeste et al. [51]. Since M540 uptake increases with high membrane destabilization, this fluorochrome has been established as a marker of sperm membrane destabilization in different mammalian species [52]. Sperm were incubated with M540 (2.6 μmol/L) and YO-PRO-1 (25 nmol/L) for 10 min at 38 °C in the dark. Four populations were identified in flow cytometry dot-plots as a result of the combination of both fluorochromes: (i) non-viable sperm with low membrane lipid disorder (M540^−^/YO-PRO-1^+^); (ii) non-viable sperm with high membrane lipid disorder (M540^+^/YO-PRO-1^+^); (iii) viable sperm with low membrane lipid disorder (M540^−^/YO-PRO-1^−^); and (iv) viable sperm with high membrane lipid disorder (M540^+^/YO-PRO-1^−^). Membrane lipid disorder was assessed as the percentage of viable, red-stained sperm with high membrane lipid disorder (M540^+^/YO-PRO-1^−^). Data were not compensated.

#### 4.4.3. Intracellular Levels of Calcium (Fluo3-AM/PI)

Evaluation of intracellular calcium levels was conducted following a protocol modified from Harrison et al. [53]. Samples were incubated with Fluo3-AM (1 μmol/L) and PI (12 μmol/L) for 10 min at 38 °C in the dark. The following four populations were distinguished in the dot-plots: (i) viable sperm with low levels of intracellular calcium (Fluo3^−^/PI^−^); (ii) viable sperm with high levels of intracellular calcium (Fluo3^+^/PI^−^); (iii) non-viable sperm with low levels of intracellular calcium (Fluo3^−^/PI^+^); and (iv) non-viable sperm with high levels of intracellular calcium (Fluo3^+^/PI^+^). The population of viable sperm with high intracellular calcium levels was used to assess intracellular calcium. Compensations for Fluo3 spill over into the FL-3 channel (2.45%) and for PI into the FL-1 channel (28.72%) were performed.

#### 4.4.4. Mitochondrial Membrane Potential (JC-1)

Evaluation of mitochondrial membrane potential (MMP) was determined through JC-1 staining, following the procedure set by Ortega-Ferrusola et al. [54] with few modifications. Samples were incubated with JC-1 (0.3 μmol/L) at 38 °C in the dark for 30 min. High MMP causes JC-1 aggregates (JC-1_agg_) that emit orange fluorescence, which is detected by FL-2. Otherwise, when MMP is low, JC-1 forms monomers (JC-1_mon_), emitting green fluorescence, which is collected through the FL-1 filter. Flow cytometry dot-plots allowed identifying the following three different populations: (i) green-stained sperm with low MMP (JC-1_mon_); (ii) orange-stained sperm with high MMP (JC-1_agg_); and (iii) double-stained sperm in both green and orange, which indicates sperm with heterogeneous mitochondria. Only orange-stained populations with high MMP were considered to assess the MMP.

#### 4.4.5. Acrosome Membrane Integrity (PNA-FITC/PI)

The integrity of acrosome membrane was assessed with the PNA-FITC fluorochrome, following the procedure of Nagy et al. [55] with minor modifications. Peanut agglutinin (PNA) lectin binds the sugar moieties, which are uniquely present in the inner leaflet of the outer acrosomal membrane [56]. Briefly, samples were incubated with PNA-FITC (2.5 μg/mL) for 5 min at 38 °C in the dark, and next, with PI (12 μmol/mL) for 5 min at 38 °C. Since sperm were not previously permeabilized, the following four populations were identified: (i) viable membrane-intact sperm (PNA-FITC^−^/PI^−^); (ii) viable sperm with a damaged plasma membrane (PNA-FITC^+^/PI^−^); (iii) non-viable sperm with a damaged plasma membrane and fully lost outer acrosome membrane (PNA-FITC^−^/PI^+^); and (iv) non-viable sperm with a damaged plasma membrane that presented an outer acrosome membrane that could not be fully intact (PNA-FITC^+^/PI^+^). Viable sperm with an intact acrosomal membrane (PNA-FITC^−^/PI^−^) were used to evaluate acrosome integrity. Compensation of PNA-FITC spill over into the PI channel (2.45%) was applied. 

#### 4.4.6. Intracellular O_2_^−^ Levels (HE/YO-PRO-1)

Evaluation of intracellular superoxide (O_2_^−^) levels in sperm was performed by co-staining with HE and YO-PRO-1, according to the protocol described by Guthrie and Welch [57]. Sperm were incubated with HE (4 μmol/L) and YO-PRO-1 (40 nmol/L) for 20 min at 38 °C in the dark. Green fluorescence from YO-PRO-1 was detected by FL-1 and the oxidation of HE to ethidium (E^+^) by O_2_^−^ was detected as red fluorescence through FL-3. The following four different populations were distinguished: (i) non-viable sperm with high O_2_^−^ levels (E^+^/YO-PRO-1^+^); (ii) non-viable sperm with low O_2_^−^ levels (E^−^/YO-PRO-1^+^); (iii) viable sperm with high O_2_^−^ levels (E^+^/YO-PRO-1^−^); and (iv) viable sperm with low O_2_^−^ levels (E^−^/YO-PRO-1^−^).

#### 4.4.7. Intracellular H_2_O_2_ Levels (H_2_DCFDA/PI)

Assessment of intracellular peroxides (H_2_O_2_) levels in sperm was performed following a protocol modified from Guthrie and Welch [57]. Sperm samples were incubated with H_2_DCFDA (10 μmol/L) for 30 min at 38 °C in the dark. Further, samples were stained with PI (12 μmol/L) for 10 min at 38 °C in the dark. The oxidation of H_2_DCFDA to DCF^+^ by H_2_O_2_ was collected by FL-1 as green fluorescence, and the red fluorescence from PI was detected by FL-3. The following four sperm populations were distinguished in dot-plots: (i) non-viable sperm with high H_2_O_2_ levels (DCF^+^/PI^+^); (ii) non-viable sperm with low H_2_O_2_ levels (DCF^−^/PI^+^); (iii) viable sperm with high H_2_O_2_ levels (DCF^+^/PI^−^); and (iv) viable sperm with low H_2_O_2_ levels (DCF^−^/PI^−^). The population of viable sperm with high H_2_O_2_ levels (DCF^+^/PI^−^) was used to assess H_2_O_2_ levels. 

### 4.5. Immunoblotting

In order to perform total protein extraction, pellets were resuspended in lysis buffer (xTractor™ Buffer; Takara Bio, Mountain View, CA, USA), supplemented with a commercial proteases inhibitor cocktail (1:100, *v*:*v*; ref. P8340) and 0.1 M phenyl-methane-sulfonylfluoride (PMSF). Samples were incubated at 4 °C for 30 min in agitation and then sonicated three times (5 × 1 s pulses at 20 KHz, 1 s pause between individual pulses). Thereafter, samples were centrifuged at 10,000× *g* for 15 min at 4 °C and supernatants were collected for subsequent protein quantification. Protein quantification was carried out in duplicate using a detergent compatible (DC) method (ref. 5000116, BioRad, Hercules, CA, USA). 

A total of 10 μg of protein was mixed with 4× Laemmli reducing buffer containing 5% β-mercaptoethanol (BioRad). Protein samples and molecular weight marker (Precision Plus Protein All Blue Standards, Bio-Rad) were incubated at 95 °C for 5 min and then loaded onto gradient commercial SDS-PAGE gels (8–16% Mini-Protean TGX Stain-Free gels; BioRad). Gels were run at 20 mA and 120–150 V using an electrophoretic system (IEF Cell Protean System, Bio-Rad). Subsequently, separated proteins were transferred onto polyvinylidene fluoride membranes (Immobilion-P; Millipore, Darmstadt, Germany) using a Trans-Blot^®^ Turbo™ device (BioRad).

Membranes were incubated overnight at 4 °C in agitation with a blocking solution consisting of 10 mM Tris (Panreac, Barcelona, Spain), 150 mM NaCl (LabKem, Barcelona, Spain), 0.05% Tween20 (Panreac, Barcelona, Spain), and 5% BSA (pH adjusted at 7.3; Roche Diagnostics, S.L.; Basel, Switzerland). Afterwards, membranes were incubated with a primary anti-PARK7 antibody (ref. LS-C353340, LifeSpan BioSciences, Seattle, WA, USA) diluted 1:5,000 (*v*:*v*) in blocking solution for 1 h at room temperature in agitation. Membranes were then washed five times with TBS1×-Tween20 (10 mM Tris, 150 mM NaCl, and 0.05% Tween20), 5 min each, and incubated with a secondary goat anti-rabbit antibody conjugated with horseradish peroxidase (ref. P0448, Agilent, Santa Clara, CA, USA) diluted 1:10,000 (*v:v*) in blocking solution for 1 h at room temperature in agitation. Subsequently, membranes were rinsed eight times with TBS1×-Tween20. Reactive bands were visualized with a chemiluminescent substrate (Immobilion^TM^ Western Detection Reagents, Millipore, Darmstadt, Germany) and blots were scanned with the G:BOX Chemi XL 1.4 device (SynGene). Next, membranes were washed twice with a stripping solution containing 0.02 mM Glycine, 0.1% SDS and 1% Tween20 (pH adjusted at 2.2) for 10 min. This was followed by four washes with TBS1×-Tween20 (10, 10, 5 and 5 min). Membranes were blocked and then incubated with a primary anti-α-tubulin antibody (ref. 05-829, Millipore) diluted at 1:100,000 (*v*:*v*) in blocking solution, for 1 h at room temperature with agitation. Thereafter, membranes were rinsed three times with TBS1×-Tween20 and incubated with a secondary rabbit anti-mouse antibody conjugated with horseradish peroxidase (ref. P0260, Agilent, Santa Clara, CA, USA) diluted 1:150,000 (*v*:*v*) in blocking solution, for 1 h at room temperature in agitation. Further, membranes were washed five times with TBS1×-Tween20, reactive bands were visualized with a chemiluminescent substrate (Immobilion^TM^ Western Detection Reagents, Millipore, Darmstadt, Germany) and blots were scanned with G:BOX Chemi XL 1.4 (SynGene).

The specificity of the primary anti-PARK7 antibody was also assessed by incubation with the corresponding PARK7-blocking peptide (ref. LS-E42782, LifeSpan BioSciences) concentrated 20 times in excess in relation to the primary antibody. The intensity of protein bands was quantified using Quantity One 1-D Analysis Software (BioRad), evaluating two technical replicates per sample. Quantifications of PARK7 protein were normalized using α-tubulin of each lane, and the corresponding mean ± SD of ratios were calculated.

### 4.6. Immunofluorescence

Immunofluorescence was performed to determine the localization of PARK7 in sperm along the incubation period (0, 120, 240, 250, and 300 min). To this end, samples collected for immunofluorescence were centrifuged at 600× *g* for 5 min to remove the CM, and sperm pellets were resuspended in PBS. Thereafter, sperm cells were fixed through incubation with 2% paraformaldehyde (*v*:*v*) for 30 min at room temperature, and were further washed twice with PBS. After fixation, sperm were placed and sedimented onto slides for 1 h at room temperature, and then permeabilized through incubation in 1% Triton X-100 in PBS for 1 h at room temperature. After permeabilization, antigen unmasking was performed exposing sperm to an acidic Tyrode’s solution for 20 s. Acid was then neutralized through washing three times with neutralization buffer, following Kashir et al. [58]. Slides were washed three times with PBS, and subsequently incubated with blocking solution (5% BSA in PBS) for 1 h at room temperature. Primary anti-PARK7 antibody (ref. LS-C353340, LifeSpan BioSciences) was diluted 1:25 (*v*:*v*) in 5% BSA in PBS (*v*:*v*) and incubated with slides for 1 h at room temperature. Subsequently, samples were incubated with a secondary antibody goat anti-rabbit Alexa Fluor^TM^ Plus 488 (ref. A32731, Invitrogen, USA) diluted 1:50 in 5% BSA in PBS (*v*:*v*) for 60 min in the dark at room temperature. Slides were then washed five times with PBS and mounted with 10 μL of ProLong^TM^ Glass Antifade Mountant with NucBlue^TM^ (Hoechst 33342; ref. P36985, Invitrogen) in the dark.

The specificity of the primary anti-PARK7 antibody was also assessed by incubation with the corresponding PARK7-blocking peptide (ref. LS-E42782, LifeSpan BioSciences) concentrated 20 times in excess in relation to the primary antibody.

Finally, all slides were observed under a confocal laser-scanning microscope (CLSM, Nikon A1R; Nikon Corp., Tokyo, Japan). To localize the nuclei stained with Hoechst 33342, samples were excited at 405 nm, whereas the localization of PARK7 was performed at an excitation wavelength of 496 nm.

### 4.7. Statistical Analyses

Results were analyzed using a statistical package (IBM for Windows 27.0; Armonk, NY, USA). Data were first checked for normal distribution (Shapiro–Wilk test) and homogeneity of variances (Levene test). When data did not fit with parametric assumptions, they were linearly transformed with arcsin √x. A two-step cluster analysis (likelihood distance and Bayesian information criterion) using the relative PARK7 content at 0 min was used to set two separate groups (sperm samples with (a) high and (b) low PARK7). Based on that classification, two groups of four samples each were identified. Following this, sperm quality and functionality parameters of these groups were compared with a linear mixed model followed by post hoc Sidak test for pair-wise comparisons (within subjects factor: incubation time; between subjects factor: sperm samples with high and low relative PARK7 content). When, even after linear transformation, data did not fit with parametric assumptions, Scheirer-Ray-Hare and Wilcoxon tests were used as alternatives. The level of significance was set at *p* < 0.05 and the confidence level was established at 95%.

## 5. Conclusions

Considering the results obtained in the present study, it can be concluded that sperm with higher levels of PARK7 at the onset of the experiment display a more progressive capacitation process with less sperm mortality. This denotes a better resilience to ROS levels, which are needed for sperm to undergo capacitation. Thus, in this context, PARK7 could play a role in preventing mammalian sperm from undergoing premature capacitation and degenerative acrosomal exocytosis. However, additional studies are needed to better understand the antioxidant properties of PARK7 during the capacitation of mammalian sperm. On the other hand, looking upon PARK7 rat homologue has been surmising to play a role in egg penetration and membrane fusion [18], performing IVF assays in other species or using knockout models could be an interesting focus for future studies. Finally, as this study was conducted in an animal model, further research using in vitro capacitated human sperm is warranted.

## Figures and Tables

**Figure 1 ijms-22-10804-f001:**
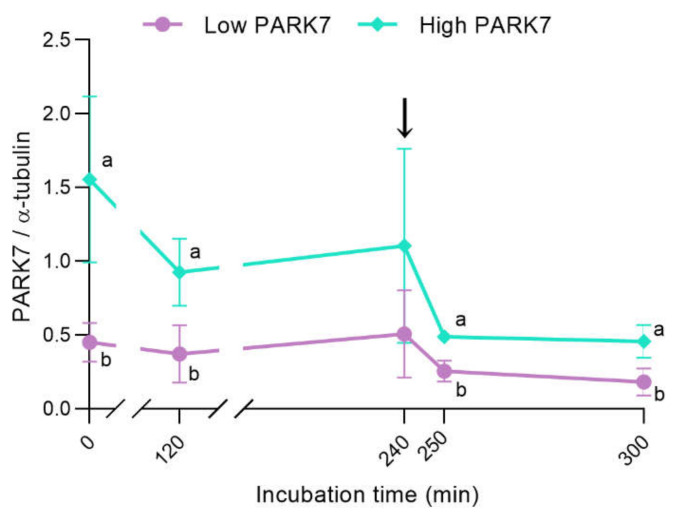
Relative PARK7 content (standardized against α-tubulin levels) during in vitro capacitation and acrosomal exocytosis induced by progesterone. The black arrow indicates the addition of progesterone at a final concentration of 10 μg/mL (240 min). Results are shown as the mean of PARK7/α-tubulin ratio ± SD. Different letters indicate significant differences (*p* < 0.05) between sperm samples with high and low PARK7 content.

**Figure 2 ijms-22-10804-f002:**
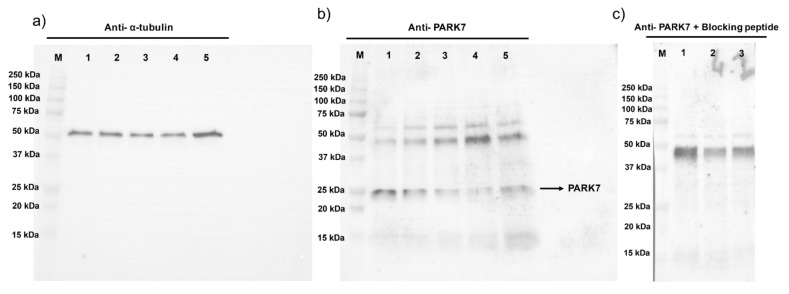
Representative immunoblots for α-tubulin (**a**), PARK7 (**b**), and peptide competition assay (anti-PARK7 + blocking peptide; (**c**). Lanes (M): protein ladder; (1) 0 min; (2) 120 min; (3) 240 min; (4) 250 min; and (5) 300 min of incubation.

**Figure 3 ijms-22-10804-f003:**
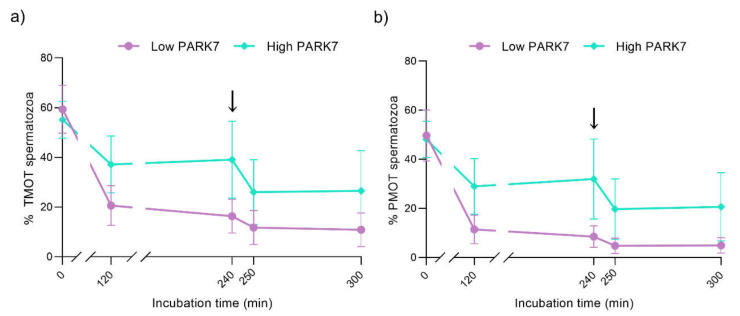
Percentages of total (**a**) and progressively (**b**) motile sperm during in vitro capacitation and acrosomal exocytosis induced by progesterone (300 min). The black arrows indicate the addition of progesterone at a final concentration of 10 μg/mL (240 min). Data are represented as means ± SD. Abbreviations: TMOT, total motility; PMOT, progressive motility.

**Figure 4 ijms-22-10804-f004:**
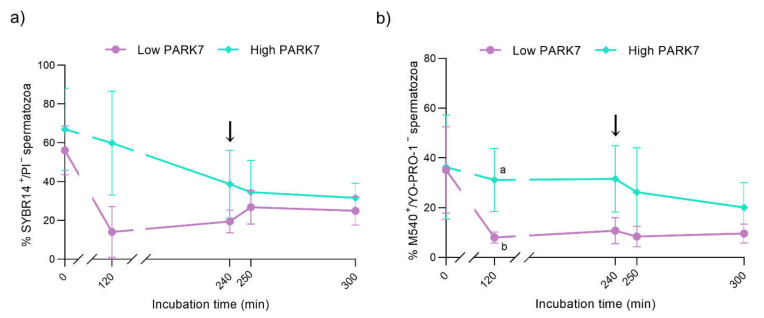
Percentages of viable sperm (SYBR14^+^/PI^−^ population) (**a**) and viable sperm with high lipid membrane disorder (M540+/YO-PRO-1- population) (**b**) during in vitro capacitation and acrosomal exocytosis induced by progesterone (300 min). The black arrows indicate the addition of progesterone at a final concentration of 10 μg/mL (240 min). Data are represented as means ± SD. Different letters indicate significant differences (*p* < 0.05) between sperm samples containing high and low relative levels of PARK7.

**Figure 5 ijms-22-10804-f005:**
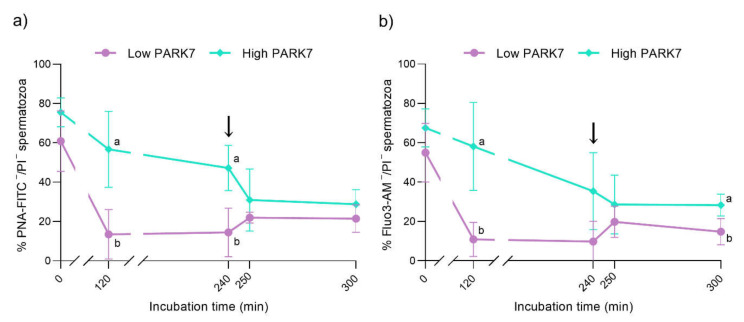
Percentages of viable sperm with an intact acrosome membrane (PNA^−^FITC-/PI^−^ population) (**a**) and viable sperm with low intracellular calcium levels (Fluo3-/PI- population) (**b**) during in vitro capacitation and acrosomal exocytosis induced by progesterone (300 min). The black arrows indicate the addition of progesterone at a final concentration of 10 μg/mL (240 min). Data are represented as means ± SD. Different letters indicate significant differences (*p* < 0.05) between sperm samples containing high and low relative levels of PARK7.

**Figure 6 ijms-22-10804-f006:**
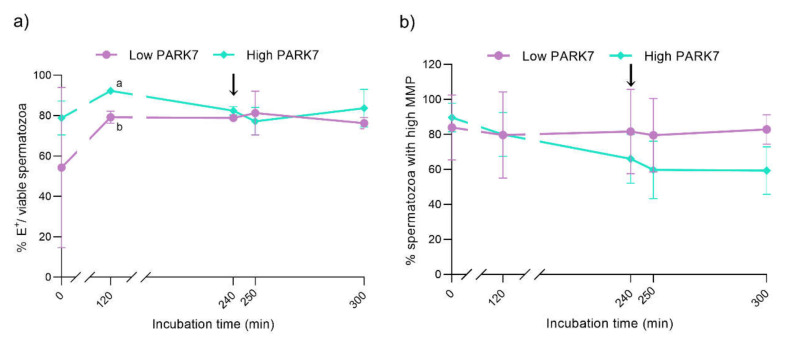
Percentages of sperm with high intracellular O_2_^−^ levels in relation to viable sperm (E^+^/viable sperm) (**a**) and sperm exhibiting high mitochondrial membrane potential (MMP) (**b**) during in vitro capacitation and acrosomal exocytosis induced by progesterone (300 min). The black arrows indicate the addition of progesterone at a final concentration of 10 μg/mL (240 min). Data are represented as means ± SD. Different letters indicate significant differences (*p* < 0.05) between sperm samples containing high and low relative levels of PARK7.

**Figure 7 ijms-22-10804-f007:**
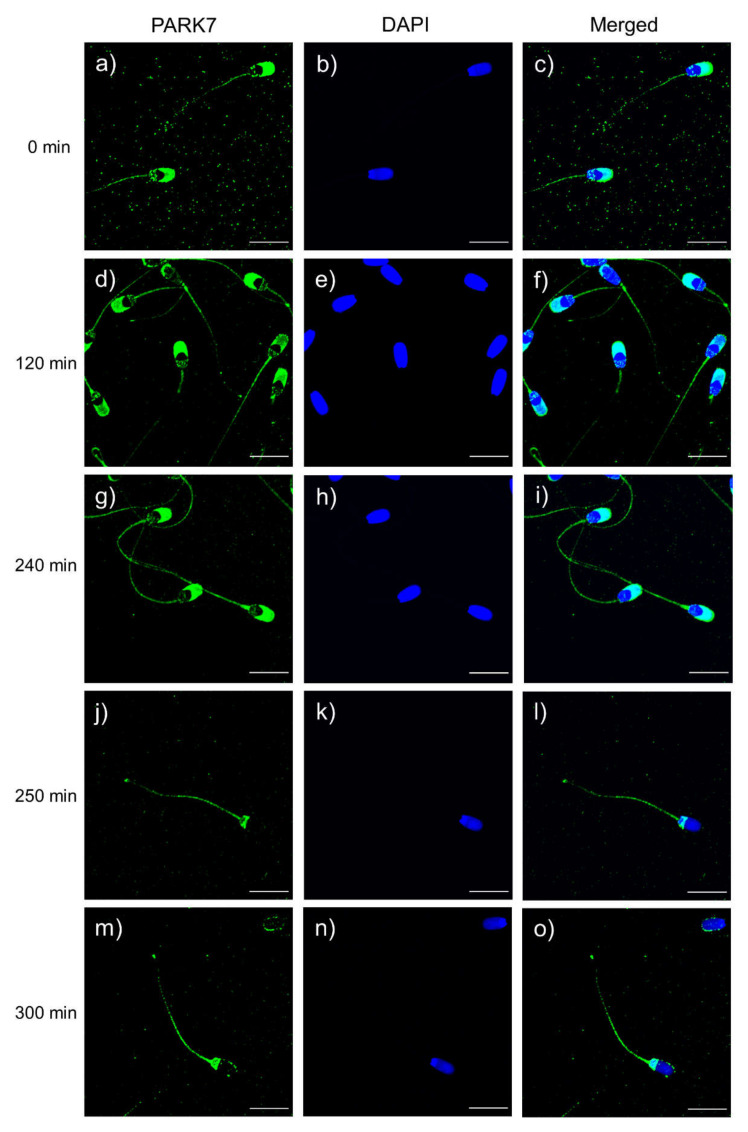
Immunolocalization of PARK7 protein at each relevant time point during in vitro sperm capacitation and acrosomal exocytosis: 0 min (**a**–**c**); 120 min (**d**–**f**); 240 min (**g**–**i**); 250 min (**j**–**l**); and 300 min (**m**–**o**). PARK7 fluorescence is shown in green, and nuclei are shown in blue (DAPI). Scale bar: 13.2 μm.

**Figure 8 ijms-22-10804-f008:**
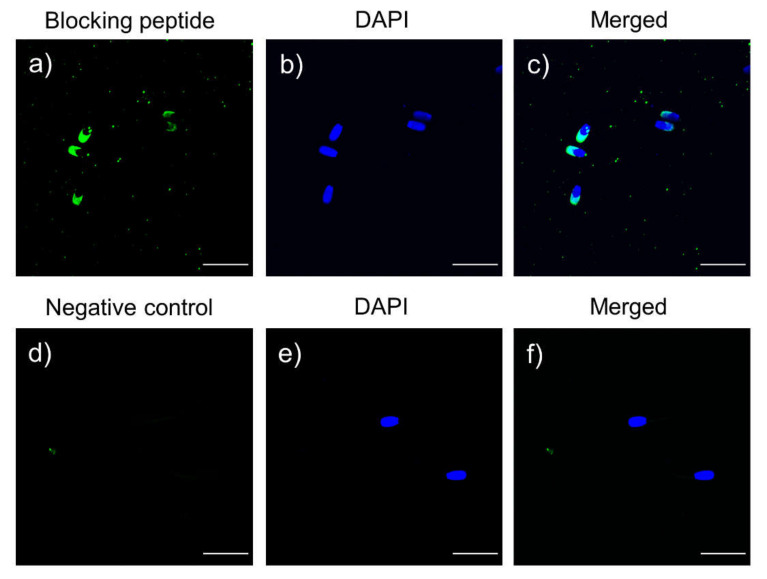
Immunofluorescence of the peptide competition assay for the anti-PARK7 antibody (**a**–**c**) and negative control (**d**–**f**). Green fluorescence indicates unspecific PARK7 staining, and nuclei are shown in blue (DAPI). Scale bar: 20 μm.

**Table 1 ijms-22-10804-t001:** Values of kinematic parameters of two groups of sperm samples, clustered on the basis of their relative PARK7 content at 0 min, during in vitro capacitation and progesterone-induced acrosomal exocytosis (0, 120, 240, 250, 300 min). Results are shown as means ± SD. VCL: curvilinear velocity; VAP: average path velocity; VSL: straight-line velocity; ALH: amplitude of lateral head displacement; BCF: beat cross frequency; LIN: linearity; WOB: wobble coefficient; STR: straightness.

		Incubation Time				
		0 min	120 min	240 min	250 min	300 min
VCL (μm/s)	Low PARK7	119.52 ± 15.09	76.15 ± 14.81	67.58 ± 4.59	65.99 ± 6.50	60.98 ± 11.27
	High PARK7	85.24 ± 35.07	77.30 ± 16.43	78.79 ± 8.14	84.73 ± 9.96	77.69 ± 7.49
VSL (μm/s)	Low PARK7	81.33 ± 16.15	42.75 ± 13.69	34.58 ± 14.31	31.09 ± 7.95	27.65 ± 11.34
	High PARK7	55.55 ± 30.51	43.37 ± 12.79	50.34 ± 6.14	55.83 ± 14.50	45.56 ± 13.55
VAP (μm/s)	Low PARK7	95.54 ± 16.06	55.03 ± 13.70	44.99 ± 12.07	42.59 ± 8.39	38.37 ± 12.66
	High PARK7	63.62 ± 34.78	51.67 ± 15.08	58.14 ± 7.50	63.39 ± 14.17	53.76 ± 12.33
LIN (%)	Low PARK7	67.53 ± 6.21	54.76 ± 8.91	50.30 ± 18.59	46.81 ± 8.89	43.36 ± 14.15
	High PARK7	62.81 ± 8.27	55.74 ± 10.33	63.97 ± 6.18	65.13 ± 9.19	57.77 ± 11.60
STR (%)	Low PARK7	84.69 ± 2.90 ^a^	75.97 ± 7.09 ^a^	74.23 ± 13.42 ^a^	71.41 ± 6.16 ^a^	68.56 ± 11.46 ^a^
	High PARK7	87.03 ± 0.46 ^a^	83.82 ± 3.70 ^a^	86.67 ± 3.80 ^a^	87.55 ± 3.87 ^b^	83.75 ± 6.63 ^a^
WOB (%)	Low PARK7	79.57 ± 4.84	71.49 ± 5.46	65.94 ± 14.16	64.34 ± 8.50	61.53 ± 11.69
	High PARK7	72.09 ± 9.24	66.26 ± 9.86	73.72 ± 4.80	74.15 ± 7.61	68.51 ± 8. 90
ALH (μm)	Low PARK7	3.71 ± 0.27 ^a^	2.57 ± 0.40 ^a^	2.23 ± 0.18 ^a^	2.52 ± 0.24 ^a^	2.23 ± 0.65 ^a^
	High PARK7	2.90 ± 0.61 ^a^	2.87 ± 0.42 ^a^	2.68 ± 0.14 ^b^	2.82 ± 0.03 ^a^	2.78 ± 0.06 ^a^
BCF (Hz)	Low PARK7	9.94 ± 0.52	8.08 ± 1.71	6.89 ± 2.16	6.53 ± 1.77	6.50 ± 2.03
	High PARK7	10.59 ± 0.44	9.88 ± 1.07	9.88 ± 0.70	9.65 ± 1.47	9.67 ± 1.21

^a,b^ indicate significant differences (*p* < 0.05) between sperm samples containing high or low levels of PARK7.

## Data Availability

The data presented in this study are available in the article and Appendix A.

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
