# Peer review of "Parkinson Disease Protein 7 (PARK7) Is Related to the Ability of Mammalian Sperm to Undergo In Vitro Capacitation"

_ijms, 2021, doi:10.3390/ijms221910804_

Round 1

Reviewer 1 Report

Topic is relevant, generally clearly presented.

The most interesting part of the manuscript is the classification of sperm samples based on their relative PARK7 content. At the beginning of the experiment (0 min), two different groups of sperm samples have been distinguished, one with high and other with low levels of PARK7, and then all of sperm parameters have been compared between sperm samples with high and low PARK7 levels.

Although this is described in point 2.7 (Statistical analysis), it would be interesting and facilitate the understanding of the manuscript, if there were a separate point in the Material and Methods that explained this classification in detail, as it happens in the Results section.

line 593 there seems to be an extra space

Author Response

REVIEWER 1:

Topic is relevant, generally clearly presented.

The most interesting part of the manuscript is the classification of sperm samples based on their relative PARK7 content. At the beginning of the experiment (0 min), two different groups of sperm samples have been distinguished, one with high and other with low levels of PARK7, and then all of sperm parameters have been compared between sperm samples with high and low PARK7 levels.

Although this is described in point 2.7 (Statistical analysis), it would be interesting and facilitate the understanding of the manuscript, if there were a separate point in the Material and Methods that explained this classification in detail, as it happens in the Results section.

We thank the reviewer for their comments, which really improve the quality of the manuscript. As suggested, we included a separate point in Material and Methods explaining the classification in detail.

Line 593 there seems to be an extra space.

Extra space has been removed.

Reviewer 2 Report

ijms-1374662

This reviewer is very familiar with Dr. Yeste’s group, and they have been publishing high-quality data with carefully designed experiments. Such is the case with their present study. This reviewer does not have any major concerns, the MS has all the necessary attributes of a high standard paper. I have minor concerns/recommendations only that should improve the comprehensibility of the study. This reviewer recommends having the MS reviewed by a native speaker as there is some awkward syntax and the use of numerous incorrect prepositions. This reviewer recommends the study for publication after my suggestions are implemented in the text.

Abstract

  1. L26-8: rewrite the sentence … higher lipid disorder, lower intracellular calcium levels, and higher superoxide levels when compared to sperm samples with lower PARK7.
  2. L30-2: add to the sentence: (Further studies are required to elucidate…) … and the relation between PARK7 and sperm motility in boars.

Introduction

  1. L59: please, change the main tail piece to the principal piece

Materials and Methods

  1. L92: Saint Louis, MO
  2. L94: Please, specify that boars were used for routine AI and provide AI success rate if available.
  3. L110: Why did the authors decide to pool 4 seminal doses when they had enough sperm, to begin with, from every boar? Was the pooling random or according to a certain algorithm? Please, specify in the MS.
  4. L112: It is a standard procedure to remove decapacitating factors prior to IVC either by Percoll centrifugation or just multiple washes in non-capacitation medium. Why did the authors decide to omit this step?
  5. L117: The symbol for kilo is small k, therefore mOsm/kg
  6. L122: Elaborate in the text why the authors find these time points relevant so that the reader does not need to assume.
  7. L126: Please, change: Samples for protein extraction were washed twice with (please add, pH=?) at 300xg…
  8. L143: Please, add a reference where the reader can look up the definitions of the said parameters.
  9. L150: Please, specify the pH of PBS used.
  10. L225: Consider changing: Peanut agglutinin (PNA) lectin binds to the sugar moieties that are uniquely present on the outer acrosomal membrane.
  11. L259: what do the authors mean by 1% protease inhibitors, as in 1X concentrated? This reviewer suggests using the catalog # and manufacturer of the cocktail.
  12. L261 and elsewhere: with agitation
  13. L261: how long the pulses lasted, how long was the pause between individual pulses? Please, specify in the MS.
  14. L264: Please, state the catalog # of the protein quantification reagent/kit
  15. L266: Incubated at 95°C, water boils at 100°C unless boiled on Puigmal
  16. L270: polyvinylidene fluoride
  17. L331-42: Please, state the confidence level used in the statistical analyses.

Results

  1. L345: Please, state that the relative content of PARK7 was determined by WB so that the reader does not have to guess.
  2. L345-7: Please, specify the number of samples for both high and low relative PARK7 content.
  3. Fig 2C: Please, show the entire blot in the MS or supplementary material.
  4. L368-9: This line seems to be in direct contradiction with the previous line. Please, correct or remove it.
  5. L401: Fig.4, how do the authors explain the rise in sperm viability in Low PARK7 from 120 mins to 250 min? It is not that the dead sperm would suddenly become viable, right?
  6. L465: The authors describe the detection of H2O2 levels in section 2.4.7 of the M&M, but the results do not contain such data. Either add the results or remove the detection of H2O2 from the M&M.

Discussion

  1. L511: …to achieve capacitated status alt. to become capacitated.
  2. L514-5: associated with…
  3. L530: … on the surface
  4. L531: From the ICC, it seems that it localizes throughout the whole midpiece
  5. L543-4: non-viable sperm with high membrane lipid disorder were
  6. L565: detrimental effects on sperm
  7. L578: The results only show superoxide levels but not peroxide levels. Please, edit accordingly.
  8. L585-8: I would disagree, there were either significant differences (P<0.05) or tendencies (0.05<P<0.1) in multiple kinematic parameters. Based on the results, the possibility of regulation of motility is plausible and cannot be ruled out. Further studies including more animals are necessary to establish a clear correlation between motility and PARK7 in boars.

Supplementary material

  1. Add a sentence that supplementary materials contain Fig S1 to S4. Supplementary figure legends should be presented in supplementary materials only.

Author Response

REVIEWER 2:

This reviewer is very familiar with Dr. Yeste’s group, and they have been publishing high-quality data with carefully designed experiments. Such is the case with their present study. This reviewer does not have any major concerns, the MS has all the necessary attributes of a high standard paper. I have minor concerns/recommendations only that should improve the comprehensibility of the study. This reviewer recommends having the MS reviewed by a native speaker as there is some awkward syntax and the use of numerous incorrect prepositions. This reviewer recommends the study for publication after my suggestions are implemented in the text.

We appreciate the positive comments from the Reviewer. The paper has been proofread to correct spelling/grammar mistakes. We also appreciate the reviewer’s suggestions, which have all been considered.  

Abstract

L26-8: rewrite the sentence … higher lipid disorder, lower intracellular calcium levels, and higher superoxide levels when compared to sperm samples with lower PARK7.

Changes have been made as requested.

L30-2: add to the sentence: (Further studies are required to elucidate…) … and the relation between PARK7 and sperm motility in boars.

Changes have been made as requested.

Introduction

L59: please, change the main tail piece to the principal piece

Changes have been made as requested.

Materials and Methods

L92: Saint Louis, MO

Changes have been made as requested.

L94: Please, specify that boars were used for routine AI and provide AI success rate if available.

The farm from which these samples were obtained told us that all AI boars were fertile. However, they could not provide the success rates for all these boars, since most of their samples were sold to other insemination centres and customers did not send the data. Thus, we have revised the Manuscript and clarified this point.

L110: Why did the authors decide to pool 4 seminal doses when they had enough sperm, to begin with, from every boar? Was the pooling random or according to a certain algorithm? Please, specify in the MS.

We thank the reviewer to address this interesting question. Indeed, this is an important point to clarify. The authors decided to randomly pool the ejaculates in order to avoid heterogeneity/variability between individuals. As suggested, this point has been clarified in the MS.

L112: It is a standard procedure to remove decapacitating factors prior to IVC either by Percoll centrifugation or just multiple washes in non-capacitation medium. Why did the authors decide to omit this step?

Actually, we previously washed sperm samples in PBS to remove seminal plasma factors (decapacitation factors), as described in Yeste et al. (Andrology 2015; 3: 729-747). However, and thanks to the reviewer, we have now realized that this was not described in the Manuscript. We have thus revised the M&M section and clarified that this step was performed.

L117: The symbol for kilo is small k, therefore mOsm/kg

Changes have been made as requested.

L122: Elaborate in the text why the authors find these time points relevant so that the reader does not need to assume.

Changes have been made as requested.

L126: Please, change: Samples for protein extraction were washed twice with (please add, pH=?) at 300xg…

Changes have been made as requested.

L143: Please, add a reference where the reader can look up the definitions of the said parameters.

A reference has been added as requested.

L150: Please, specify the pH of PBS used.

Changes have been made as requested.

L225: Consider changing: Peanut agglutinin (PNA) lectin binds to the sugar moieties that are uniquely present on the outer acrosomal membrane.

Changes have been made as requested.

L259: what do the authors mean by 1% protease inhibitors, as in 1X concentrated? This reviewer suggests using the catalog # and manufacturer of the cocktail.

We have changed the expression ‘1%’ to ‘1:100 (v:v)’ and the catalogue number has been added. As indicated at the beginning of Material and Method section, the protease inhibitor cocktail was purchased from Sigma-Aldrich.

L261 and elsewhere: with agitation

Changes have been made as requested.

L261: how long the pulses lasted, how long was the pause between individual pulses? Please, specify in the MS.

This information has been specified in the MS body as requested.

L264: Please, state the catalog # of the protein quantification reagent/kit

Changes have been made as requested.

L266: Incubated at 95°C, water boils at 100°C unless boiled on Puigmal

Changes have been made as requested.

L270: polyvinylidene fluoride

Changes have been made as requested.

L331-42: Please, state the confidence level used in the statistical analyses.

Changes have been made as requested.

Results

L345: Please, state that the relative content of PARK7 was determined by WB so that the reader does not have to guess.

Changes have been made as requested.

L345-7: Please, specify the number of samples for both high and low relative PARK7 content.

Changes have been made as requested.

Fig 2C: Please, show the entire blot in the MS or supplementary material.

Figure 2C shows the entire blot as only a part of a membrane was used to perform the peptide competition assay. Please see attached the entire image (as the image is not visible, it is included in the rebuttal letter).

L368-9: This line seems to be in direct contradiction with the previous line. Please, correct or remove it.

Changes have been made as requested.

L401: Fig.4, how do the authors explain the rise in sperm viability in Low PARK7 from 120 mins to 250 min? It is not that the dead sperm would suddenly become viable, right?

The mean percentage of viable sperm at 120 min was 14.08 (SD = 13.11), whereas it was 26.80 (SD = 8.77) at 250 min. However, the difference observed in sperm viability between these two time point was not statistically significant (p = 0.952). Thus, such an apparent rise in sperm viability is probably an artefact resulting from the high variability observed, especially at 120 min. Since the most important point here is that no significant differences between 120 and 250 min were observed, we did not pay much attention to this apparent difference.

L465: The authors describe the detection of H2O2 levels in section 2.4.7 of the M&M, but the results do not contain such data. Either add the results or remove the detection of H2O2 from the M&M.

Results of H2O2 and O2 levels were described in the same section (Section 3.7. Intracellular ROS levels). No significant differences between samples with different relative content of PARK7 regarding H2O2 levels were observed.

Discussion

L511: …to achieve capacitated status alt. to become capacitated.

Changes have been made as requested.

L514-5: associated with…

Changes have been made as requested.

L530: … on the surface

Changes have been made as requested.

L531: From the ICC, it seems that it localizes throughout the whole midpiece

Changes have been made as requested.

L543-4: non-viable sperm with high membrane lipid disorder were

Changes have been made as requested.

L565: detrimental effects on sperm

Changes have been made as requested.

L578: The results only show superoxide levels but not peroxide levels. Please, edit accordingly.

Please see our previous response about superoxide and peroxide levels.

L585-8: I would disagree, there were either significant differences (P<0.05) or tendencies (0.05<P<0.1) in multiple kinematic parameters. Based on the results, the possibility of regulation of motility is plausible and cannot be ruled out. Further studies including more animals are necessary to establish a clear correlation between motility and PARK7 in boars.

We thank the reviewer for this comment. Taking their consideration into account, the sentence has been revised.

Supplementary material

Add a sentence that supplementary materials contain Fig S1 to S4. Supplementary figure legends should be presented in supplementary materials only.

Changes have been made as requested.

Reviewer 3 Report

This is an interesting study that investigated the role of PARK7 in sperm function. Two groups of sperm based on relative PARK7 expression are identified. The authors found that sperm with high PARK7 expression have higher viability and intact acrosome. They also showed high lipid disorder, low intracellular calcium and high superoxide levels. This study reports novel roles of PARK7 for sperm function.

Here are some comments

Line 55: have been reported to

Line 58: is localized to the surface of sperm head,

Line 60: flagellar movement

Line 61: on the surface of

Line 69: under oxidative stress

Line 70: associated with, translocate from to

Line 72: protective for

Line 73: downregulated not under-expressed

Line 76: detrimental effects on

Line 79: referred to them as

Line 346-347: Please add standard deviation for mean PARK7 expression in sperm cells.

Line 414: with low relative PARK7, please use the word relative consistently, same line 429, 432, 448, 463, 459, 476, please check throughout the results

Line 447: was not

The number of figures is high, it is better to combine them together and reduce the total number of figures

Part 3.4, can you specify which specific part of sperm membrane?

Part 3.9: regarding PARK7 localization in sperm, it is shown from immunofluorescence that that it is also expressed in sperm midpiece that includes mitochondria as well, so please add this part to line 488.

According to the peptide competition assay, Park7 is not expressed in acrosomal region since it disappeared, how do you account for the part of the results that indicates that relative park7 expression correlates with acrosome membrane integrity?

It is better to put the results part about PARK7 expression before other results, laying out hypothesis for investigating this gene in sperm function.

What is known about PARK7 protein and mRNA expression in other organs?

What is known about regulation of PARK7 mRNA expression in sperm?

What is known about male fertility of PARK7 homozygous or knockout models?

Is PARK7 required for egg penetration? Have you performed IVF assay for sperm samples with high and low PARK7 relative expression?

Line 514: associated with

Line 518: change underexpressed to downregulated  

Line 522: remove the before human

Author Response

REVIEWER 3:

This is an interesting study that investigated the role of PARK7 in sperm function. Two groups of sperm based on relative PARK7 expression are identified. The authors found that sperm with high PARK7 expression have higher viability and intact acrosome. They also showed high lipid disorder, low intracellular calcium and high superoxide levels. This study reports novel roles of PARK7 for sperm function.

Here are some comments

Line 55: have been reported to

Changes have been made as requested.

Line 58: is localized to the surface of sperm head,

Changes have been made as requested.

Line 60: flagellar movement

Changes have been made as requested.

Line 61: on the surface of

Changes have been made as requested.

Line 69: under oxidative stress

Changes have been made as requested.

Line 70: associated with, translocate from to

Changes have been made as requested.

Line 72: protective for

Changes have been made as requested.

Line 73: downregulated not under-expressed

Changes have been made as requested.

Line 76: detrimental effects on

Changes have been made as requested.

Line 79: referred to them as

Changes have been made as requested.

Line 346-347: Please add standard deviation for mean PARK7 expression in sperm cells.

Changes have been made as requested.

Line 414: with low relative PARK7, please use the word relative consistently, same line 429, 432, 448, 463, 459, 476, please check throughout the results

Changes have been made as requested.

Line 447: was not

Changes have been made as requested.

The number of figures is high, it is better to combine them together and reduce the total number of figures

The number of figures has been reduced as suggested.

Part 3.4, can you specify which specific part of sperm membrane?

Examining M540 staining under a fluorescence microscope is required to determine whether sperm membrane is destabilised and which specific parts are affected. As reported by Harrison et al. (1996), changes in lipid architecture in response to bicarbonate exposure can be observed by an increase in M540 staining over the whole surface area of sperm cell membrane, and specially in sperm head and mid-piece.

Part 3.9: regarding PARK7 localization in sperm, it is shown from immunofluorescence that that it is also expressed in sperm midpiece that includes mitochondria as well, so please add this part to line 488.

Changes have been made as requested.

According to the peptide competition assay, Park7 is not expressed in acrosomal region since it disappeared, how do you account for the part of the results that indicates that relative park7 expression correlates with acrosome membrane integrity?

Since sperm need to achieve the capacitated status to undergo acrosome reaction, effects observed in the acrosome integrity are a direct consequence of the capacitated status of sperm. PARK7 seems to be related to the sperm ability to withstand oxidative stress generated during capacitation, and therefore, could play a role preventing sperm from premature capacitation. Thus, sperm containing lower relative amount of PARK7 may be susceptible to a premature capacitation and acrosome reaction, regardless the fact that PARK7 does not localise in the acrosome. This hypothesis does not need PARK7 to be located in the acrosomal region.

It is better to put the results part about PARK7 expression before other results, laying out hypothesis for investigating this gene in sperm function.

We thank the reviewer for this suggestion, which really improves the discussion of the results and facilitates their comprehension to the reader. We have moved the section of WB results as requested.

What is known about PARK7 protein and mRNA expression in other organs?

The study of Yoshida et al. (2003) detected the presence of PARK7 protein by immunohistochemical staining in human epididymis and testis, whereas Bandopadhyay et al. (2004) confirmed the expression of PARK7 in different regions of the brain. Cloning and expression analysis of porcine PARK7 revealed the expression of this protein in a variety of tissues, including cerebellum, frontal cortex, pituitary gland, thyroid gland, liver, spleen, kidney, lung, muscle, stomach, fat tissue, and heart. However, PARK7 has been studied into more detail concerning Parkinson disease and neurodegenerative disorders, considering the different functions associated to this protein, including antioxidant activity, transcriptional regulation, chaperone activity, and protein degradation (reviewed by Hijioka et al. 2017). We added part of this information at the beginning of the discussion section.

What is known about regulation of PARK7 mRNA expression in sperm?

Unfortunately, there is no literature available about this issue.

What is known about male fertility of PARK7 homozygous or knockout models?

Unfortunately, there is no literature available about this issue.

Is PARK7 required for egg penetration? Have you performed IVF assay for sperm samples with high and low PARK7 relative expression?

To the best of our knowledge, the study of Klinefelter et al. (2002) was the only one which investigated the relationship between PARK7 and fertilising ability. Incubation of rat cauda epididymal sperm with SP22 (rat homologue of PARK7) antibodies inhibited fertilisation in vitro, and seemed to be associated with a decrease in the number of sperm bound to the oocyte. Thus, Klinefelter et al. (2002) suggested that SP22 may play a role in both zona pellucida penetration and membrane fusion. However, neither this putative function has been investigated in other species, nor we performed any IVF assay in order to study the correlation with the relative content of PARK7. Taking the reviewer’s comment into consideration, we have now included a reference to this potential future subject of research in the conclusion section.

Line 514: associated with

Changes have been made as requested.

Line 518: change underexpressed to downregulated 

Changes have been made as requested.

Line 522: remove the before human

Changes have been made as requested.